# Differentiation of Endometriomas from Ovarian Hemorrhagic Cysts at Magnetic Resonance: The Role of Texture Analysis

**DOI:** 10.3390/medicina56100487

**Published:** 2020-09-23

**Authors:** Roxana-Adelina Lupean, Paul-Andrei Ștefan, Csaba Csutak, Andrei Lebovici, Andrei Mihai Măluțan, Rareş Buiga, Carmen Stanca Melincovici, Carmen Mihaela Mihu

**Affiliations:** 1Histology, Morphological Sciences Department, “Iuliu Hațieganu” University of Medicine and Pharmacy, Louis Pasteur Street, Number 4, Cluj-Napoca, 400349 Cluj, Romania; roxanalupean92@gmail.com (R.-A.L.); carmen.melincovici@umfcluj.ro (C.S.M.); carmenmihu@umfcluj.ro (C.M.M.); 2Obstetrics and Gynecology Clinic “Dominic Stanca”, County Emergency Hospital, 21 Decembrie 1989 Boulevard, Number 55, Cluj-Napoca, 400094 Cluj, Romania; amalutan@umfcluj.ro; 3Anatomy and Embryology, Morphological Sciences Department, “Iuliu Haţieganu” University of Medicine and Pharmacy, Victor Babeș Street, Number 8, Cluj-Napoca, 400012 Cluj, Romania; 4Radiology and Imaging Department, County Emergency Hospital, Cluj-Napoca, Clinicilor Street, Number 5, Cluj-Napoca, 400006 Cluj, Romania; csutakcsaba@yahoo.com (C.C.); andrei1079@yahoo.com (A.L.); 5Radiology, Surgical Specialties Department, “Iuliu Haţieganu” University of Medicine and Pharmacy, Clinicilor Street, Number 3-5, Cluj-Napoca, 400006 Cluj, Romania; 6Obstetrics and Gynecology Clinic II, Mother and Child Department, “Iuliu Hațieganu” University of Medicine and Pharmacy, 21 Decembrie 1989 Boulevard, Number 55, Cluj-Napoca, 400094 Cluj, Romania; 7Pathological Anatomy, Morphological Sciences Department, “Iuliu Hațieganu” University of Medicine and Pharmacy, Victor Babeș Street, Number 8, Cluj-Napoca, 400012 Cluj, Romania; raresbuiga@yahoo.fr

**Keywords:** ovarian cysts, endometriosis, endometrioma, magnetic resonance imaging (MRI), texture analysis

## Abstract

*Background and Objectives*: To assess ovarian cysts with texture analysis (TA) in magnetic resonance (MRI) images for establishing a differentiation criterion for endometriomas and functional hemorrhagic cysts (HCs) that could potentially outperform their classic MRI diagnostic features. *Materials and Methods:* Forty-three patients with known ovarian cysts who underwent MRI were retrospectively included (endometriomas, *n* = 29; HCs, *n* = 14). TA was performed using dedicated software based on T2-weighted images, by incorporating the whole lesions in a three-dimensional region of interest. The most discriminative texture features were highlighted by three selection methods (Fisher, probability of classification error and average correlation coefficients, and mutual information). The absolute values of these parameters were compared through univariate, multivariate, and receiver operating characteristic analyses. The ability of the two classic diagnostic signs (“T2 shading” and “T2 dark spots”) to diagnose endometriomas was assessed by quantifying their sensitivity (Se) and specificity (Sp), following their conventional assessment on T1-and T2-weighted images by two radiologists. *Results*: The diagnostic power of the one texture parameter that was an independent predictor of endometriomas (entropy, 75% Se and 100% Sp) and of the predictive model composed of all parameters that showed statistically significant results at the univariate analysis (100% Se, 100% Sp) outperformed the ones shown by the classic MRI endometrioma features (“T2 shading”, 75.86% Se and 35.71% Sp; “T2 dark spots”, 55.17% Se and 64.29% Sp). *Conclusion*: Whole-lesion MRI TA has the potential to offer a superior discrimination criterion between endometriomas and HCs compared to the classic evaluation of the two lesions’ MRI signal behaviors.

## 1. Introduction

Endometriosis is defined as the presence of functional endometrial tissue outside the uterine cavity [1]. Endometriomas are the most common form of endometriosis, being produced by repeated hemorrhage from an ectopic endometrium located in the ovaries [2]. They present in the form of cysts resulting from cyclic bleeding over months [3], being responsible for almost 17% of infertility cases in women [4]. On the other hand, functional hemorrhagic ovarian cysts (HCs) occur primarily during ovulation [5] as a result of bleeding in a luteal or follicular cyst, and are often asymptomatic [6].

Correctly identifying the two lesions is crucial to eschew unneeded surgery [7], but also because endometriomas are a marker of the presence of other endometriotic lesions at the pelvic and intestinal level, which can cause a series of complications [8]. Thus, the imaging differentiation of the two entities can influence the course of both medical and surgical treatment [9]. Transvaginal ultrasonography (TVUS) is the primary diagnostic tool in the diagnosis of endometriomas. In most cases, this technique can provide sufficient information for adequate preoperative planning [10]. Other imaging investigations, such as magnetic resonance imaging (MRI), are performed in selected cases, according to the TVUS results and the severity of symptoms. The pelvic MRI examination allows complete lesion mapping, showing a high detection rate for both anterior and especially posterior lesions, where TVUS shows a decreased sensitivity [11]. 

Since first described by Nishimura and colleagues [12] in 1987, the “T2 shading” sign has become a classic MRI feature of endometriomas. This sign describes an adnexal cyst that exhibits a high signal for T1-weighted images (T1WI) and a low signal for T2-weighed images (T2WI) [13]. Initially, it was thought that this appearance was very characteristic of endometriomas, with Togashi et al. [14] reporting a high sensitivity (98%) and specificity (96%). The signal decrease in T2WI can be partially attributed to the high concentration of blood products and proteins usually found in this type of lesion [13]. However, because sometimes HCs may contain a large quantity of blood, this phenomenon can also manifest, mimicking the “T2 shading” sign [2]. Recent studies that aimed to discriminate endometriomas from other adnexal lesions complicated by bleeding (including HCs) observed that the “T2 shading” sign has a lower diagnostic ability than was previously thought (68–93% sensitivity, 45–93% specificity) [2,13,15]. The reported shortcomings of this diagnostic sign evoke the subjective nature of its assessment [16] and its occurrence in other adnexal cysts complicated by hemorrhage [3,17,18]. A relatively newly introduced MRI feature of endometriomas is the “T2 dark spots” that represent imaging manifestations of chronic retracted blood clots [2]. Although blood clots can also form in simple hemorrhagic cysts, they do not have time to lose free water and build up hemosiderin concentrations (due to the short-lived nature of these cysts), and therefore they do not produce focal T2 shortening [2]. Corwin et al. [2] claimed that these “dark spots” have a high specificity (93%) but a low sensitivity (36%) for distinguishing endometriomas from non-endometriotic hemorrhagic cystic lesions. A common limitation of both signs is given primarily by the absence of defined imaging criteria for their identification.

The pathological analysis shows important differences in cellularity and physical and biochemical features between endometriomas and HCs [19,20]. It is theorized that these features can also have an impact on the pixel intensity and distribution within MRI images, but that their influence is too subtle to be assessed by the routine evaluation of these examinations. Textures represent the intrinsic and intuitive properties of surfaces such as roughness, granulation, and regularity. Texture analysis (TA) is a technique based on the extraction and processing of image-specific parameters that provides an objective description of image content by quantifying the distribution patterns and intensity of the pixels [21]. Due to the numerous applications involving medical images, this method has been integrated as a core component of computer-aided diagnosis [22].

In the present research, TA was used to quantify the imaging features of endometriomas and functional hemorrhagic cysts. The aim was to determine if texture parameters can provide additional information that can be used for distinguishing between the two entities, thus exploring the possibility of a new non-invasive diagnosis approach for ovarian hemorrhagic lesions that could outperform the classic MRI diagnosis signs.

## 2. Materials and Methods

### 2.1. Patients

This Health Insurance Portability and Accountability Act for a compliant, single-institution, and retrospective pilot study has been approved by the institutional review board (ethics committee of the “Iuliu Hațieganu” University of Medicine and Pharmacy Cluj-Napoca; registration number, 50; date, 11.03.2019), and a waiver consent was obtained owing to its retrospective nature. In our radiology database, reports of pelvic MRI scans were searched from January 2017 to February 2020 using the keywords: “endometriosis”, “endometrioma/s”, “hemorrhagic cyst/s”, and “ovarian cyst/s”. The original search yielded 162 reports. Each report was then analyzed and those studies in which the keyword in the report did not refer to the presence of an ovarian hemorrhagic cyst were excluded (*n* = 47). The medical records of the remaining 115 patients were retrieved from the archive of our healthcare unit and investigated for disease-related data. The other exclusion criteria were: other benign or malignant ovarian lesions complicated by bleeding (*n* = 8), lesions with a diameter of less than 15 mm (*n* = 19), the presence of artifacts within the T2WI (*n* = 6), patients without a final clinical or pathological diagnosis of the lesion (*n* = 21), or patients who did not undergo gynecological follow-up (*n* = 18).

### 2.2. Pathological Analysis

In the case of endometriomas, 26 patients underwent surgery followed by the pathological confirmation of the disease (mean time from MRI examination to surgery, 22.6 days; range, 12–43 days). Three lesions were diagnosed by clinical and imaging methods, followed by gynecologists via TVUS for at least four months (mean follow-up period, 168.4 days; range 129–231 days). All the HCs represented incidental findings on MRI, with the patients being investigated for another pelvic pathology. Eight cysts were surgically removed through oophorectomy and underwent pathological analysis along with the underlying disease. Six HCs underwent gynecological follow-up, and their remission in time was noted at the following US examinations (time from MRI to remission as seen on TVUS: mean, 72.3; range, 44–83 days).

The surgical samples were fixed overnight in 10% buffered formalin and then were automatically embedded in paraffin, using tissue processors, according to the usual protocol of the pathological anatomy laboratory. The resulting blocks were sectioned at 5 μm, and the slides were stained with hematoxylin and eosin. All the slides were examined by a pathologist with experience in gynecological pathology. Additional immunohistochemical staining was rarely necessary to confirm the presence of endometrial tissue in the histological sections. Immunohistochemical staining was used for pan-cytokeratin AE1/AE3 and CK7 to identify glandular epithelial cells, for CD10 to identify the endometrial stromal component, and sometimes for Ki67 to assess the proliferation index of lesions. Following this diagnostic workflow, in all the cases examined it was possible to histologically distinguish between the two entities.

### 2.3. MRI Protocol

All the MRI scans were performed on the same unit (SIGNA™ Explorer, 1.5 Tesla, Waukesha, WIS, USA) under a routine pelvic protocol. All the subjects were instructed not to move and to maintain expiratory apnea according to instructions. The protocol included: axial T1-weighted fast spin-echo (FSE) sequence with a large field of view (FOV), sagittal T2-weighted (T2W) with periodically rotated overlapping parallel lines with enhanced reconstruction (PROPELLER), oblique axial T2W fast relaxation fast spin-echo (FR-FSE) at high resolution (HR), oblique coronal T2 FR-FSE HR, axial diffusion-weighted sequences (DWI) with 3 b values (50,400, 1000 s/mm^2^), and contrast-enhanced T1 fat-suppressed. The acquisition protocol varied in the four-year-range from which the examinations were selected, but each protocol consisted of an oblique axial T2 FR-FSE HR sequence, which was the only sequence used for texture analysis. The parameters of this sequence were: repetition time, 4556 ms; echo time, 109 ms; bandwidth, 83.333 Hz/pixel; FOV, 240 mm; matrix, 320 × 320; section thickness, 4 mm; and slice gap, 0.4 mm. No routine antiperistaltic drugs were used, but instead a narrow band around the abdomen was applied to diminish the intestinal peristalsis.

### 2.4. Image Interpretation

On a dedicated workstation (General Electric, Advantage workstation, 4.7 edition), all the examinations were reviewed by one radiologist and one gynecologist (C.S.M. and A.M.M.), who were aware of the patients’ final pathological and clinical outcomes. When multiple bleeding cysts were observed within the same examination, the images were cross-referenced with the pathological and TVUS results and other medical data to ensure the selection of lesions that were previously documented. Only one lesion from each patient was selected and marked.

The “T2 shading” and the “T2 dark spots sign” of the selected lesions were evaluated by two radiologists (C.C. and A.L., each with at least 10 years of experience in pelvic MRI), blinded to the clinical and pathological outcome, and the results were recorded by consensus. The “T2 shading” was assessed according to the protocol proposed by Dias et al. [13], being considered when an ovarian cyst showed a complete or partial loss of signal intensity between T1 and T2WI. The “T2 dark spots” were assessed as specified in the research by Corwin et al. [2], according to which they present on T2WI as hypointense well-defined foci within the cysts but not within the wall. 

### 2.5. Texture and Statistical Analysis

The radiomics approach consisted of three steps: image segmentation using regions of interest, feature extraction, and feature selection. Images of the previously selected lesions from the oblique axial T2W sequence were anonymized and retrieved in Digital Imaging and Communications in Medicine (DICOM) format. A third radiologist (P.A.S.), blinded to the final diagnosis, imported each image set into a texture analysis software, MaZda version 5 [23]. The image segmentation step consisted of incorporating each cystic lesion into a three-dimensional (3D) region of interest (ROI) (Figure 1). A semi-automatic level-set technique was used for the definition and positioning of each ROI using gradient and geometry coordinates. Since this technique does not require the manual delineation of the structure of interest contours, the inter-and intra-observer reproducibility was not assessed in this study. By applying a limitation of dynamics to μ ± 3σ (μ = gray-level mean; σ = gray-level standard deviation), the gray level was normalized to reduce the influence of contrast and brightness that could affect the true image textures [24].

The feature extraction step was automatically performed by the MaZda program. Over 300 texture parameters were generated from the analysis of each ROI, parameters which originated from the grey-level histogram, co-occurrence matrix, run-length matrix, absolute gradient, Haar wavelet transformation, and autoregressive model. Since such a large amount of data is difficult to be analyzed by conventional statistical methods, the MaZda software allows its reduction by the use of selection methods. Each of these methods provides a set of 10 features that are best suited for discriminating between the vector groups. Three reduction techniques were applied based on the Fisher coefficients (F, the ratio of between-class to within-class variance), the probability of classification error and average correlation coefficients (POE+ACC), and mutual information (MI) [25].

A univariate analysis test (Mann–Whitney U) was conducted to compare the absolute values of each texture parameter highlighted by the three selection methods. The statistical significance level was set at a *p*-value of below 0.0016 after applying the Bonferroni correction (which implied dividing the standard 0.05 value to 31 variables; 30 variables were represented by the parameters highlighted by each selection method, plus one variable corresponding to the patients’ age). A receiver operating characteristic (ROC) analysis was performed, with the calculation of the area under the curve (AUC) with 95% confidence intervals (CIs) for the parameters showing statistically significant results at the univariate analysis. Furtherly, a multiple regression analysis using an “enter” input model was conducted to identify which of the texture parameters that showed statistically significant results in the univariate analysis are also independent predictors of endometriomas. The coefficient of determination (R-squared) was computed, and the diagnostic value of the prediction model was evaluated using ROC analysis. 

The ability of the two classic diagnostic signs (“T2 shading” and “T2 dark spots”) to diagnose endometriomas was assessed by quantifying their accuracy (expressed as the percentage of correctly classified lesions), sensitivity (true positive rate), and specificity (true negative rate). Statistical analysis was performed using a commercially available dedicated software, MedCalc version 14.8.1 (MedCalc Software, Mariakerke, Belgium).

The feature name generated by the MaZda software contains abbreviations of feature characteristics produced by the extraction algorithm. The outermost symbol from the left indicates the first imaging processing procedure. The first letter indicates the color channel (“C” implies that a black and grey image was computed, “R” identities the red color channel). The second symbol stands for image normalization (N), after which follows the encoding for the method used—in this case, “S” represents image normalization using the limitation of dynamics to μ ± 3σ. The following number indicates that the feature was quantized to use that number of bits per pixel. The direction is coded using letters: H (horizontal), V (vertical), Z (45°), and N (135°). The next group of letters identities the extraction algorithm (e.g., Wav, Haar wavelet transformation). The feature name is usually the last group of letters (LngREmph, long-run emphasis; ShrtREmp, short-run emphasis; En, energy) [26]. 

## 3. Results

Of the 162 patients who were referred to our department during the study period, 43 were included in the study after applying the inclusion and exclusion criteria (mean age, 34.6; range, 21–39 years). The subjects were divided according to the final pathological and/or clinical diagnosis in the endometrioma (*n* = 29) and HCs group (*n* = 14). The conventional imaging evaluation showed that 22 endometriomas and 9 HCs demonstrated the “T2 shading” sign. Sixteen endometriomas and 5 HCs demonstrated “T2 dark spots”. The diagnostic power of these signs in identifying endometriomas is shown in Table 1. 

Two variations of the wavelet energy parameter (WavEn) computed from the low-low (LL) frequency band within the first (s-1) and second (s-2) decomposition levels were highlighted by all three classification methods. Alongside these parameters, seven variations of entropy, three of the angular second moment (AngScMom), one of the sum entropy (SumEntrp), and one of the histogram’s variance (Variance) showed statistically significant results in the univariate analysis (Table 2). The ROC results of the parameters that showed statistically significant results in the univariate analysis are displayed in Table 3 and Figure 2. 

The multiple regression analysis indicated that CV2S6Entropy was the only parameter independently associated with the presence of endometriomas (Table 4). The prediction model consisting of all the parameters that showed statistically significant results in the univariate analysis was able to identify endometriomas with a sensitivity of 100% (CI, 85.8–100%) and a specificity of 100% (CI, 71.5–100%). The AUC displayed by the prediction model was 1 (CI, 0.9–1) and the significance level was < 0.0001 (Figure 2).

## 4. Discussion

The MRI appearance of endometriomas has been extensively evaluated before. Unfortunately, very few studies have aimed to differentiate them from HCs or even from other adnexal lesions complicated by bleeding (Table 5). 

The difference between the two entities has also been investigated through more advanced MRI techniques, such as diffusion-weighted imaging, via the measurement of the apparent diffusion coefficient (ADC). Two studies investigated the restriction of water molecules inside endometriomas and HCs, and although these measurements had a very good diagnostic ability, the absolute measured values for each entity were opposed in the two studies [2,29]. Although the “T2 dark spot” sign was described as more characteristic of endometriomas [2] the sensitivity and specificity reported in previous studies [2,3] showed contradictive results. Additionally, in our study only almost half of these lesions exhibited this MRI signal behavior.

The previous reports show that for every imaging feature there is a wide range of diagnostic values, which partially may be due to physicians assessing these features only qualitatively. Moreover, there are no criteria defined about how wide the T2 shortening must be to represent an accurate diagnostic sign, apart from the fact that it is only subjectively assessed. These limitations create the need for more information to be extracted from medical images and to develop quantitative benchmarks for the diagnosis of these lesions.

Texture analysis includes a wide variety of feature categories, each reflecting certain aspects of pixel intensity and patterns. The wavelet decomposition represents a multiresolution technique that intends to transform images into a representation in which both spatial and frequency information is present, which can be accomplished by the multiple direction filtering of an image. This technique decomposes the image information in the horizontal, vertical, and diagonal directions, each direction being described using specific texture parameters. The values of the wavelet energy parameter reflect the variations in pixel intensity within an image [30]. Our results show lower values of both WaveEn parameters for the HCs, most likely because the HCs have a more uniform content, thus providing lower signal variation rates. 

Entropy measures the degree of the disorder among pixels within an image, being inversely correlated with uniformity [31]. A high degree of randomness within an image produces high entropy levels, while sum entropy represents an indicator of the complexity of pixel values distribution within an image [32]. Our results showed higher median values of all the entropy and sum entropy parameters in endometriomas, probably due to their multiple types of cell populations and heterogeneous biochemical components compared to the relative plainness of functional cyst content [20]. Additionally, a variation of entropy (CV2S6Entropy) was the only one reported as being an independent predictor of endometriomas (*p* = 0.0014)

The variance parameter represents the sum of the squares of the differences between the intensity of the central pixel and its neighbors [33]. The angular second moment measures the local uniformity of an image, having high values when the grey level distribution has either a constant or a periodic form [34]. Our results showed higher values of the AngScMom parameter and lower values of variance for the HCs group, indicating a more uniform content for these lesions. Considering these observations, it can be concluded that all the above-mentioned texture analysis results are predictors of irregularity within endometriomas, possibly due to the reflection of some of the histological features of these lesions in their MRI imaging appearance. 

The Sum Average (SumAverg) parameter measures the average sum of gray levels [35]. All six selected variations of this parameter showed lower values for endometriomas than for HCs (probably because the premiums contained more blood products that produced a higher T2 shortening), but the differentiation between the two groups was not statistically significant. Additionally, previous studies evaluating the role of this parameter in medical imaging analysis observed that it possesses low diagnostic value [36]. The first-order histogram is one of the most common statistical methods for image feature computation. The histogram does not consider the spatial relations between the pixels, reflecting only the value of their intensity [37]. Several such histogram parameters are represented by the percentiles. The percentile number (n) is the point at which n% of the pixel values that form the histogram are found to the left [38,39]. This implies that 90% and 99% of the pixels within images were distributed under lower values for HCs, but again the differences were not statistically significant. The Run Length Non-Uniformity (RLNonUni) measures the non-uniformity of the grey levels or the length of the homogeneous runs [40]. This parameter recorded higher values for endometriomas than for HCs, probably due to the variable content of the premiums. The correlation (Correlat) parameter measures image linearity, its values increasing when an image contains linear structures [41]. Again, this parameter recorded higher values for endometriomas, but the difference between the two groups was not statistically significant (*p* = 0.4).

It is important to evaluate every piece of information found in the standard MRI sequences, rather than adding more sequences or even contrast enhancement. We successfully demonstrated that, by applying TA on a standard T2 FR-FSE HR sequence, endometriomas could be accurately discriminated from hemorrhagic cysts. If it could be further validated, this technique could reduce the MRI acquisition time and overall examination cost and avoid unnecessary contrast administration, which would be truly beneficial to patients. 

It is well documented that endometriomas’ content changes with their evolution over time. Thus, old lesions contain a much larger amount of bilirubin and ferritin, a higher overall density, and especially a higher collagen content than early lesions. With collagen accumulation and fibrosis production, the endometriomas become lighter in color and resistant to hormonal treatment [42]. Normally, only microscopic evaluation can make a clear differentiation between the endometriomas of different evolutionary stages [43]. Thus, it is possible that, in the future, TA could provide a non-invasive characterization of the cystic content, and through this make an accurate prediction about the response rates of hormonal treatments. Although this was not the objective of the present study, our results encourage future research that aims to differentiate between evolutionary stages and the treatment response of these lesions based on TA parameters.

It was previously documented that endometriomas’ fluid content has the characteristics of a non-Newtonian liquid, with inhomogeneous viscosity within the same lesion [42]. Additionally, because blood products may be dispersed, it was justified to integrate the whole lesion in a 3D ROI. Because all the examinations were performed on the same device and using the same protocol, the resulting images have a high degree of homogeneity, thus being able to counteract the variations in textural measurements produced by the use of multiple scanners or different examination protocols [22]. If validated on larger studies, this approach can be beneficial to patients with undetermined TVUS who are referred to MRI for an extensive evaluation of a hemorrhagic ovarian lesion. 

Our study had several limitations. First, due to its retrospective design it could have had selection bias, but it remains debatable whether the inclusion of both pathologically confirmed and unconfirmed lesions could be regarded as a pitfall. The diagnosis of endometriomas and functional hemorrhagic cysts is also based on clinical and paraclinical criteria, and an important part of these lesions is treated conservative [44]. Moreover, a previously published paper that evaluated the utility of the “T2 shading” sign [3] stated that the choice of including only histologically proven lesions produced a “greater selection bias”, decreasing the specificity of the reported findings. Second, being a retrospective study may have selection and verification bias regarding the selected patients and gynecological follow-up, which mainly depend on the status of the institution and referral hospital. Third, the number of subjects was relatively low, and the overall number of endometriomas exceeded almost two times the number of HCs. Another limitation may be regarded as the lack of reproducibility (including inter-and intra-observer differences) investigation, although previous research following the same method stated that, due to the semi-automatic ROI positioning, this assessment is not necessary [45]. The fact that two researchers (C.S.M. and A.M.M.) were aware of the final diagnosis can also be viewed as a limitation. However, this approach was necessary because at the time of the MRI examinations several patients had multiple lesions, and it was desired to include only the ones that were pathologically or at least clinically documented. After this stage, these investigators (C.S.M. and A.M.M.) did not intervene in any way in interpreting the images, reporting the results, or conducting the statistical analysis. Additionally, the MaZda software used in this article can be regarded as outdated, since the official version had not received improvements in more than 10 years. However, in this study, we used a newly developed Beta version of this software released in 2016 (available online: https://data.mendeley.com/datasets/dkxyrzwpzs/1) [46]. Although more modern dedicated TA software is free and commercially available, MaZda steel represents a valid TA method, since it provides one of the largest numbers of feature customization, selection, extraction, and processing methods. Additionally, it offers an intuitive interface and, thus, the possibility of being used by non-image processing specialists, such as regular physicians. The MaZda software also enables the use of several classifiers. In this paper, we preferably applied a more conventional approach to the statistical processing of the parameters due to the fact that several of these classifiers (such as the artificial neural networks) require a large amount of data for an adequate classification procedure. However, this change in statistical processing may have affected our results. To build a robust predictive model through multivariate analysis, the feature subset should contain mostly uncorrelated parameters [47]. Therefore, our selection method involving Fisher coefficients highlights parameters that, besides having a high discriminatory potential, are also well correlated with each other [25]. This may have affected the overall performance of our combined prediction model, partially altering its diagnostic value. Moreover, there are several technical factors that could have influenced our texture analysis model. Firstly, the use of a high-resolution sequence was shown to improve the texture discrimination [48]. Secondly, the study was conducted in a single institution, thus reducing the differences in texture features extracted from MRI images from different centers [49]. As mentioned before, the variations in textural features were also reduced by using examinations provided by a single scanner and performed by the same protocol [22]. Thus, it is less likely that the same high performance in discriminating the two groups of lesions could be achieved by the use of multiple protocols or multiple scanners.

## 5. Conclusions

The quantitative evaluation of MRI images through whole-lesion texture analysis has the potential to outperform classic diagnostic signs of endometriomas. Moreover, texture analysis offers a quantitative description of these lesions, which could avoid diagnostic errors due to the subjective interpretation of medical images. As the first study investigating the textural properties of ovarian hemorrhagic lesions, the results are promising and open the way for many future types of research.

## Figures and Tables

**Figure 1 medicina-56-00487-f001:**
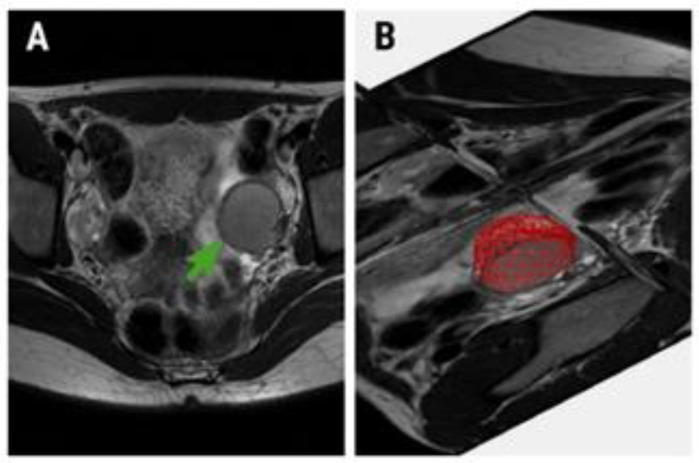
(**A**) Oblique axial T2-weighted image of a 32-year-old patient with pathologically confirmed endometrioma (green arrow) and (**B**) the region of interest (red area) used for texture analysis.

**Figure 2 medicina-56-00487-f002:**
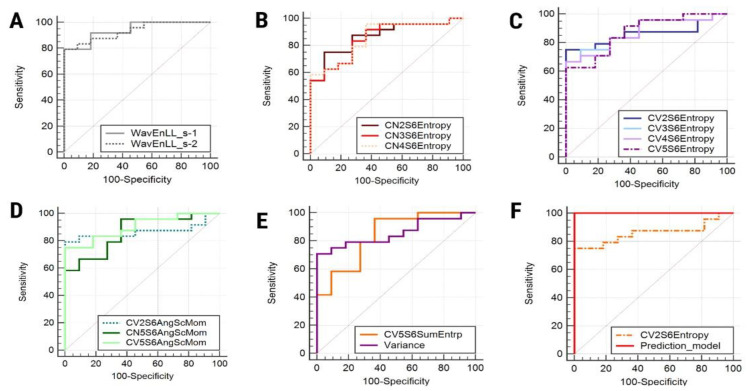
Receiver operating characteristic curve of wavelet energy variations (**A**), entropy variations computed for the 135° direction (**B**), entropy variations computed for vertical directions (**C**), angular second moment variations (**D**), sum entropy and variance (**E**), and (**F**) the entropy parameter that represents an independent predictor of endometriomas (CV2S6Entropy) and the prediction model itself. WavEn, wavelet energy; Entropy, entropy; AngScMom, angular second moment; SumEntrp, sum entropy; Variance, histogram’s variance.

**Table 1 medicina-56-00487-t001:** The diagnostic power of the classic magnetic resonance imaging (MRI) signs in the diagnosis of endometriomas. Between the brackets, values correspond to the 95% confidence interval (CI).

Diagnostic Sign	Endometriomas	HCs	Accuracy (%)	Sensitivity (%)	Specificity (%)
“T2 shading”	*n* = 22	*n* = 9	62.79 (46.73–77.02)	75.86 (56.46–89.7)	35.71 (12.76–64.86)
“T2 dark spots”	*n* = 16	*n* = 5	58.14 (42.13–72.99)	55.17 (35.69–73.55)	64.29 (35.14–87.24)

**Table 2 medicina-56-00487-t002:** The parameters highlighted by each selection method, the univariate analysis (Mann–Whitney U test) results, and the median values obtained by these parameters in the case of endometriomas and hemorrhagic cysts (HCs).

Parameter	Coefficient	*p*-Value	Endometriomas	HCs
*Fisher*				
WavEnLL_s-1	4.4737	**<0.0001**	17,748.26 (17,365.55–18,027.58)	15,355.26 (14,043.69–16,607.25)
WavEnLL_s-2	3.4413	**0.0001**	17,373.98 (16,457.74–18,525.13)	12,582.52 (10,020.4–15,096.37)
CV4S6Entropy	2.6443	**0.0006**	2.74 (2.55–2.93)	2.1 (1.87–2.55)
CV5S6Entropy	2.634	**0.0005**	2.76 (2.55–2.93)	2.09 (1.7–2.57)
CV3S6Entropy	2.4381	**0.0004**	2.66 (2.54–2.93)	2.1 (1.97–2.49)
CV5S6SumEntrp	2.4259	**0.0014**	1.79 (1.73–1.83)	1.52 (1.22–1.75)
CN3S6Entropy	2.4018	**0.001**	2.76 (2.58–2.92)	2.09 (2.01–2.56)
CN2S6Entropy	2.3572	**0.0006**	2.69 (2.53–2.92)	2.13 (2.07–2.49)
CV2S6Entropy	2.3346	**0.0007**	2.62 (2.49–2.88)	2.17 (2.06–2.4)
CN4S6Entropy	2.2939	**0.0008**	2.79 (2.52–2.92)	2.1 (1.87–2.59)
*POE + ACC*				
Perc99	0.249	0.0219	33,336 (132–33,579.5)	148 (123.5–166.7)
WavEnLL_s-1	0.2857	**<0.0001**	17,748.26 (17,365.55–18,027.58)	15,355.26 (14,043.69–16,607.25)
WavEnLL_s-2	0.34	**0.0001**	17,373.98 (16,457.74–18,525.13)	12,582.52 (10,020.4–15,096.37)
CV1S6AngScMom	0.3407	0.002	0.003 (0.002–0.006)	0.01 (0.007–0.013)
RHS6RLNonUni	0.3413	0.0025	1651.2 (410.94–2727.26)	158.82 (78.83–995.16)
Variance	0.3478	**0.0008**	2401.3 (328.5–5182.46)	120.51 (40.09–238.33)
CV5S6AngScMom	0.3483	**0.0002**	0.002 (0.001–0.003)	0.009 (0.003–0.021)
Perc90	0.3854	0.024	33,270.5 (111 to 33,500)	135 (114.5–157.25)
CN5S6AngScMom	0.397	**0.0006**	0.001 (0.001–0.003)	0.012 (0.003–0.02)
CH1S6AngScMom	0.4135	0.0032	0.003 (0.002–0.007)	0.008 (0.066–0.01_
*Mutual Information*				
CV2S6SumAverg	0.6806	0.54	64.58 (64.07–65.04)	65.83 (62.5–66.99)
CN2S6SumAverg	0.6235	0.6189	64.5 (64.16–65.02)	66.2 (62.39–67.89)
CV3S6SumAverg	0.5879	0.4996	64.56 (63.82–65.13)	66.27 (62.36–67.06)
CZ1S6SumAverg	0.5412	0.7223	64.64 (64.3–64.87)	66.19 (62.54–67.5)
CN1S6SumAverg	0.5045	0.6698	64.62 (64.22–64.9)	65.76 (62.33–67.26)
WavEnLL_s-2	0.4639	**0.0001**	17,373.98 (16,457.74–18,525.13)	12,582.52 (10,020.4–15,096.37)
WavEnLL_s-1	0.4525	**<0.0001**	17,748.26 (17,365.55–18,027.58)	15,355.26 (14,043.69–16,607.25)
CH2S6SumAverg	0.431	0.7223	64.75 (64.18–64.97)	66.31 (62.22–68.19)
CV2S6AngScMom	0.426	**0.0006**	0.002 (0.001–0.004)	0.008 (0.006–0.018
CZ5S6Correlat	0.4086	0.0252	0.586 (0.048–0.757)	0.21 (0.466–0.269)

HCs, hemorrhagic cysts; WavEn, wavelet energy; Entropy, entropy; SumEntrp, sum entropy; Perc99, 99% percentile; Perc 90, 90% percentile; AngScMom, angular second moment; RLNonUni, run-length nonuniformity; Variance, histogram’s variance; SumAverg, sum average; Correlat, correlation; POE + ACC, probability of classification error and average correlation coefficient. Values in bold are statistically significant. Between the brackets, values correspond to the interquartile range.

**Table 3 medicina-56-00487-t003:** The receiver operating characteristic analysis results of the texture parameters that showed statistically significant results at the univariate analysis for the diagnosis of endometriomas. Between the brackets, values correspond to the 95% CI.

Parameter	AUC	Sign.lvl.	J	Cut-Off	Se (%)	Sp (%)
WavEnLL_s_1	0.939 (0.804–0.992)	<0.0001	0.7917	>17,177.882	79.17 (57.8–92.9)	100 (71.5–100)
WavEnLL_s_2	0.932 (0.793–0.989)	<0.0001	0.7917	>16,321.281	79.17 (57.8–92.9)	100 (71.5–100)
CV4S6Entropy	0.867 (0.710–0.958)	<0.0001	<0.0001	>2.6335	66.67 (44.7–84.4)	100 (71.5–100)
CV5S6Entropy	0.871 (0.715–0.960)	<0.0001	0.625	>2.6625	62.5 (40.6–81.2)	100 (71.5–100)
CV3S6Entropy	0.875 (0.719–0.962)	<0.0001	0.6667	>2.5915	66.67 (44.7–84.4)	100 (71.5–100)
CV2S6Entropy	0.86 (0.701–0.954)	<0.0001	0.75	>2.4921	75 (53.3–90.2)	100 (71.5–100)
CN3S6Entropy	0.852 (0.692–0.949)	<0.0001	0.5606	>2.4254	83.33 (62.6–95.3)	72.73 (39–94)
CN2S6Entropy	0.867 (0.710–0.958)	<0.0001	0.6591	>2.5666	75 (53.3–90.2)	90.91 (58.7–99.8)
CN4S6Entropy	0.856 (0.696–0.951)	<0.0001	0.5947	>2.1539	95.83 (78.9–99.9)	63.64 (30.8–89.1)
CV5S6AngScMom	0.902 (0.753–0.976)	<0.0001	0.75	≤0.0032	75 (53.3–90.2)	100 (71.5–100)
CN5S6AngScMom	0.864 (0.705–0.956)	<0.0001	0.5947	≤0.0083	95.83 (78.9–99.9)	63.64 (30.8–89.1)
CV2S6AngScMom	0.867 (0.710–0.958)	<0.0001	0.7917	≤0.0043	79.17 (57.8–92.9)	100 (71.5–100)
CV5S6SumEntrp	0.841 (0.678–0.942)	<0.0001	0.5947	>1.5482	95.83 (78.9–99.9)	63.64 (30.8–89.1)
Variance	0.856 (0.696–0.951)	<0.0001	0.7083	>769.4818	70.83 (48.9–87.4)	100 (71.5–100)

AUC, area under the curve; Sign.lvl., significance level; J, Youden index; Se, sensitivity; Sp, specificity; WavEn, wavelet energy; Entropy, entropy; AngScMom, angular second moment; SumEntrp, sum entropy; Variance, histogram’s variance.

**Table 4 medicina-56-00487-t004:** Multivariate analysis of parameters independently associated with the presence of endometriomas.

Independent Variable	Coefficient	Standard Error	*p*-Value	VIF
CN2S6Entropy	−8.2967	4.3623	0.0717	1027.045
CN3S6Entropy	15.6427	7.5443	0.0513	3669.47
CN4S6Entropy	−8.1976	4.398	0.0771	1673.578
CN5S6AngScMom	−5.4068	6.6826	0.428	31.822
CV2S6AngScMom	93.1964	59.4064	0.1324	28.594
CV2S6Entropy	9.3644	2.5189	**0.0014**	306.535
CV3S6Entropy	−12.7996	4.4433	0.0092	1125.262
CV4S6Entropy	5.4712	6.3882	0.4019	2655.577
CV5S6AngScMom	23.9864	63.1998	0.7083	127.429
CV5S6Entropy	0.1952	4.8106	0.968	1932.615
CV5S6SumEntrp	2.0136	1.2166	0.1135	31.504
Variance	0.00002574	0.00001358	0.0727	1.521
WavEnLL_s_1	0.0001645	0.0001153	0.1693	16.494
WavEnLL_s_2	−0.00002706	0.00005094	0.6011	15.087
**Sign.lvl.**	0.0007			
**R^2^**	0.7724			
**R^2^ adjusted**	0.6131			
**M.R. Coef.**	0.8789			

VIF, Variance Inflation Factor; R^2^, coefficient of determination; R^2^ adjusted, coefficient of determination adjusted for the number of independent variables in the regression model; Sign.lvl., significance level of the multivariate analysis; M. R. Coef., multiple regression coefficient. Bold values are statistically significant.

**Table 5 medicina-56-00487-t005:** Research involving endometrioma diagnosis classic MRI features.

Imaging Feature	Author	Diagnostic Value
“T2 dark spots”	* Corwin et al. [2]	36% (19.8–51.3%) Se; 93% (83.9–100%) Sp.
	* Lee et al. [3]	95.2% Se; 14.3% Sp.
Morphological and signal features (including the “T2 shading “sign)	* Corwin et al. [2]	93% (84–100%) Se; 45% (27.8–61.9%) Sp.
	Togashi et al. [14]	90% Se; 98% Sp.
	Sugimura et al. [27]	82% Se; 91% Sp for lesions > 1 cm
		11% Se; 98% Sp for lesions < 1 cm
	Scout et al. [28]	92% Se; 91% Sp.
	Lee et al. [3]	89.8% Se; 14.2% Sp.
	* Outwater et al. [15]	68% Se; 76% Sp.
	* Dias et al. [13]	73% (56.8–85.2%) Se; 93% (89.3–95.5%) Sp.
ADC values	* Lee et al. [2]	100% Se; 92% Sp.
	* Balaban et al. [29]	77.6% Se; 76.2% Sp.

MRI, magnetic resonance imaging; Se, sensibility; Sp, specificity; ADC, apparent diffusion coefficient; * studies that aimed to differentiate endometriomas from other hemorrhagic ovarian cystic entities.

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
