# Peer review of "Differentiation of Endometriomas from Ovarian Hemorrhagic Cysts at Magnetic Resonance: The Role of Texture Analysis"

_medicina, 2020, doi:10.3390/medicina56100487_

Round 1
Reviewer 1 Report
Background:
Page 2 line 53-54: 17% of infertility cases in 53 women [4]. Reference 4 is a rather old reference (7 years) but perhaps it still is up to date.
I miss information about texture analysis and the different categories the study uses in its result. It had made the result easier to read. Some if this I found in the discussion side 9 line 285-293. And some of the information on line 294 “Entropy measures the degree of the disorder among pixels within an image” and 302-303 “The variance parameter represents the sum of the squares of the differences between the 302 intensity of the central pixel and its neighbors”.
Material and Method:
Why was oblique axial T2 FR-FSE HR sequence used for texture analysis? Was it necessary to use a T2 weighted image? Had it the best resolution? Or had it the best angulation?
Why was Mann–Whitney U used for analysis? Non parametric data? Small sample?
P-value level used for significance is often 0.05, 0.01 or 0.001 but in this study 0.0016 is used. This is rather uncommon. Why using it?
Result:
Table 2, even if it is stated in the method how p-value is calculated, it should be mentioned in the table.
As mentioned, the parameters in table 2 parameters should had been explained in the background.
Discussion:
There is two table 3.
page 9, Line 285-291 fit in the background.
Page 10, line 306 it is written "We consider", we is not an optimal word, perhaps " it was considered" 322 its again "we".
It is a small sample, but it is well explained why and it is also called “a pilot study”
Reviewer 2 Report
In my opinion, the submitted text concerning quantification of MR imaging features of endometriomas and hemorrhagic cysts by means of texture analysis based on wavelet decomposition, presents valuable, well designed, elaborated and clearly described work. Authors claim significantly better performance of their proposed method over the classic MRI diagnostics based on radiologist's vsual assessment, in terms of sensitivity and specificity, both reaching 100%. It is important that they indicate also possible limitations of their work, which may lead to such extremely good performance. It is valuable work, showing possible direction for further improvement of relevant MRI based diagnostics.
I have a few minor comments:
L148 do you mean "contiguous slice spacing" ?
L191 please explain how do you get the Bonferroni corrected p-value to be 0.0016 ? It is not clear, based on information you provided.
L256 you probably mean "sensitivity", not "sensibility" ?
While language of the text is generally correct, please check carefully for some spelling errors, examples being e.g. in L163 "withing" or in L359 "sings" which should be "within" and "signs", respectively.
